# A One-Pot Convenient RPA-CRISPR-Based Assay for *Salmonella enterica* Serovar Indiana Detection

**DOI:** 10.3390/microorganisms12030519

**Published:** 2024-03-05

**Authors:** Jiansen Gong, Di Zhang, Lixia Fu, Yongyi Dong, Kun Wu, Xinhong Dou, Chengming Wang

**Affiliations:** 1Poultry Institute, Chinese Academy of Agricultural Sciences, Yangzhou 225125, China; jjsensen@163.com (J.G.); douxinhong611@163.com (X.D.); 2Jiangsu Co-Innovation Center for the Prevention and Control of Important Animal Infectious Disease and Zoonose, Yangzhou University, Yangzhou 225009, China; 3Key Laboratory for Poultry Genetics and Breeding of Jiangsu Province, Jiangsu Institute of Poultry Sciences, Yangzhou 225125, China; shawn_zd@126.com; 4College of Animal Science and Technology, Yangzhou University, Yangzhou 225009, China; dynamicren@163.com; 5Jiangsu Animal Disease Prevention and Control Center, Nanjing 210036, China; jsyk2014@163.com (Y.D.); wukun20000@126.com (K.W.); 6Department of Pathobiology, College of Veterinary Medicine, Auburn University, Auburn, AL 36849, USA

**Keywords:** *Salmonella* Indiana, RPA, CRISPR/Cas12b, one pot, one step

## Abstract

*Salmonella enterica* serovar Indiana (*S.* Indiana) is among the most prevalent serovars of *Salmonella* and is closely associated with foodborne diseases worldwide. In this study, we combined a recombinase polymerase amplification (RPA) technique with clustered regularly interspaced short palindromic repeat (CRISPR) and CRISPR-associated (Cas) protein Cas12b (CRISPR/Cas12b)-based biosensing in a one-pot platform to develop a novel one-step identification method for *S.* Indiana infection diagnosis. The entire RPA-CRISPR/Cas12b reaction can be completed at 41 °C within 1 h without the need for specific instruments. The optimal concentrations of Cas12b and single-guide RNA (sgRNA) for the reaction were the same at 250 nM. The single-stranded DNA (ssDNA) reporter 8C-FQ (5′-/6-FAM/CCCCCCCC/BHQ1/-3′) presented the best performance in the reaction compared with the other reporters. The limit of detection (LoD) of the RPA-CRISPR/Cas12b assay was 14.4 copies per reaction. As for specificity, we successfully identified four *S.* Indiana strains among twenty-two *Salmonella* strains without any false-positive results, presenting 100% accuracy for *S.* Indiana, and no cross-reactions were observed in eight other pathogens. Moreover, a total of 109 chicken carcasses were classified by the *S.* Indiana RPA-CRISPR assay and PCR methods from three processing points, including 43 post-shedding, 35 post-evisceration, and 31 post-chilling. There were 17 *S*. Indiana-positive samples identified during the whole processing step, consisting of nine post-shedding, five post-evisceration, and three post-chilling. The corresponding *S.* Indiana-positive rates of post-shedding, post-evisceration, and post-chilling were 20.93% (9/43), 14.29% (5/35), and 9.68% (3/31), respectively. Results from the *S.* Indiana one-step RPA-CRISPR/Cas12b assay were totally in agreement with those obtained using a traditional culture method, demonstrating 100% agreement with no false-positive or false-negative results observed. Altogether, the RPA-CRISPR/Cas12b assay developed in this study represents a promising, accurate, and simple diagnostic tool for *S.* Indiana detection.

## 1. Introduction

*Salmonella*, a widely recognized foodborne pathogen with a global reach, represents a primary instigator of foodborne illness outbreaks, posing significant threats to human health and resulting in substantial economic ramifications annually [1,2]. The prevalence of *Salmonella* infections remains staggering, with over 93.8 million cases reported each year, accompanied by approximately 155,000 related fatalities [3]. The emergence of *Salmonella* enterica serovar Indiana (*S*. Indiana) dates back to its initial identification in Indiana in 1955 when it was isolated from a young girl afflicted with symptoms including vomiting, diarrhea, and fever. Since its discovery, the *S*. Indiana pathogen has been implicated in numerous infections among both humans and mammals across North America and Europe [4]. In recent years, China has witnessed a notable surge in the prevalence of *S*. Indiana, coupled with concerning levels of drug resistance [5,6].

Previous reports have indicated that all *S.* Indiana strains can be isolated from chicken industry chains [7,8], making individuals susceptible to *S*. Indiana infection through poultry products in daily life. The extensive expansion of *S.* Indiana has caused an increasing threat to global public health [9]. To maintain health and prevent the outbreak of foodborne illness caused by *S.* Indiana, a convenient method for early *S.* Indiana detection is urgently needed. 

Microbiological culturing is a traditional method of *S.* Indiana identification and is time-consuming, labor-intensive, and low-efficiency [10]. Polymerase chain reaction (PCR)-based methods have become routine assays and are currently widely applied for pathogen detection. Data collected from our previous report indicated that the LoD of the PCR method for *S.* Indiana detection is 10 pg per reaction for bacterial genomic DNA, equivalent to 100 colony-forming units (CFUs) per reaction [11]. Utilizing molecular-based methodologies, PCR-based methods can yield results with superior efficiency and specificity, enabling the rapid detection of foodborne *Salmonella*. [12,13]. However, these tools are highly dependent on professional equipment, experienced operators, and long reaction times. To address these issues, several isothermal amplification techniques have been discovered and utilized for detection, such as loop-mediated isothermal amplification (LAMP) and recombinase polymerase amplification (RPA) [14,15]. Moreover, our group has explored the LAMP method for *Salmonella* spp. detection based on targeting the *bcfD* gene, which has the advantages of rapidity, sensitivity, specificity, and practicality [16]. These methods can be rapidly applied because of the significant advantages of simple instrumentation, time saving, and simple procedures. However, great efforts are needed to improve specificity.

Recently, clustered regularly interspaced short palindromic repeats (CRISPR)/CRISPR-associated (CRISPR/Cas) systems have provided novel insights into pathogen detection. Cas effectors (such as Cas12a, Cas12b, and Cas13a) can exert collateral cleavage activity on non-target single-strand RNA and DNA after recognizing and cleaving the target sequence in the presence of specific RNA-guided nucleases [17]. Moreover, several CRISPR/Cas-based platforms have been used in nucleic acid analysis, represented by SHERLOCK (Specific High-Sensitivity Enzymatic Reporter Unlocking) [18], HOLMES (One-Hour Low-Cost Multipurpose Highly Efficient System) [19], DETECTR (DNA Endonuclease-Targeted CRISPR Trans Reporter) [20], and others. Commonly, pre-amplification and CRISPR-based detection are the two main separate steps for most CRISPR-based detection systems, which cannot totally avoid contamination from multiple operations and aerosols. Meaningfully, a developed CRISPR-based system with one pot and one step has been used for the detection of coronavirus disease 2019, which overcomes the drawbacks of the traditional CRISPR-based method [21]. However, the corresponding CRISPR-based method for *S.* Indiana identification has not yet been established.

In this study, we integrated the recombinase polymerase amplification (RPA) technique with a CRISPR/Cas12b-based system in a unified one-pot platform to establish a novel identification system (RPA-CRISPR/Cas12b) for diagnosing *S*. Indiana infections. Our findings demonstrate the significant advantages of this system, including simplicity, rapidity, accuracy, and freedom from contamination in detecting *S*. Indiana infections.

## 2. Materials and Methods

### 2.1. Reagents and Instruments

The RAA nuclear amplification kit was obtained from Qitian Biotech (B00000, Wuxi, China), AapCas12b was purchased from TOLO Biotech (32118, Shanghai, China), HOLMES ssDNA reporter (FAM) was provided from TOLO Biotech (31101, Shanghai, China), the Cas12b High Yield sgRNA Synthesis and Purification Kit was purchased from TOLO Biotech (31904, Shanghai, China), and the Digital PCR Mixture was purchased from ZHENZHUN BIO (MX0108, Shanghai, China). Nuclease-free water was purchased from Solarbio Life Sciences (R1600, Beijing, China).

The instruments used in the present study are listed as follows: real-time PCR system (SLAN-96S, HONGSHI, Shanghai, China), QuantStudio 3 real-time quantitative PCR system (QuantStudio 3, ThermoFisher, Waltham, MA, USA), QuantStudio 5 real-time quantitative PCR system (QuantStudio 5, ThermoFisher, Waltham, MA, USA), Qubit fluorescent spectrophotometer (Qubit4, ThermoFisher, Waltham, MA, USA), AccuMini Digital PCR System (AccuMini, ZHENZHUN BIO, Shanghai, China).

### 2.2. Sample Preparation

A total of 30 strains were used in this study, including 22 *Salmonella* strains (18 reference strains and 4 *S*. Indiana strains) and 8 non-*Salmonella* reference strains (Table 1). All of the reference strains were collected from the ATCC (American Type Culture Collection), NCTC (National Collection of Type Culture), CMCC (National Center for Medical Culture Collections), and CICC (China Center of Industrial Culture Collection). The origins of strains used in this study (S1105, S1467, and S1515) were recovered from aquatic product, broiler, and duck, and the detailed information can be found in our previous publication [22]. These strains were archived at −80 °C in tryptic soy broth (TSB) (Hopebiol, Qingdao, China) containing 20% glycerol. The commercial DNA isolation kit (TIANGEN, Beijing, China) was used to extract genomics DNA of *S*. Indiana according to the instructions of the manufacturer. To generate the recombinant plasmid pUC57-SI_A7P63_0910, the target gene A7P63_0910 was inserted into the pUC57 vector. The concentration of the recombinant plasmid was 19.25 ng/μL, which corresponds to a copy number of 5.36 × 10^9^ copies/μL. Then, the plasmid was used as a standard product for the RPA-CRISPR/Cas12b detection system.

### 2.3. The Principle and Workflow of the RPA-CRISPR/Cas12b Detection System for S. Indiana 

In the present research, the workflow of the RPA-CRISPR/Cas12b detection system is shown in Figure 1. In brief, a commercial DNA extraction kit was used to extract the crude genomic DNA of *S.* Indiana. The whole reaction, RPA integration with CRISPR/AapCas12b-based detection, was performed in a single reaction step at a constant temperature for *S.* Indiana nucleic acid detection. The RPA-amplified products can be recognized by the corresponding AapCas12b/sgRNA system and subsequently activate Cas12b, resulting in trans-cleavage of the reporter DNA. The cleavage of report DNA, which was labeled with fluorophore 6-FAM and quencher BHQ1, resulted in the appearance of fluorescence, then the results can then be detected in the real-time PCR fluorescence readout. The whole test can be completed within 1 h, including 15 min for rapid template preparation and 45 min for RPA-CRISPR/Cas12b detection. 

### 2.4. The Design and Selection of RPA Primers for the S. Indiana RPA-CRISPR/Cas12b Assay

Here, the A7P63_0910 (GenBank: ANF77768.1) gene was chosen as the specific target for *S.* Indiana, which is uniquely present in *S.* Indiana, but not in other *Salmonella* serovars or any non-*Salmonella* bacteria, and used as a target gene for *S.* Indiana detection [23]. A total of 7 pairs of RPA primers (named SI-1 to SI-7, seen in Table 2) were designed according to the conserved region of A7P63_0910, and the amplification products were analyzed by electrophoresis in agarose gels.

As shown in Appendix A, the amplification product of RPA reaction by using primer SI-5 presents the best quality. Then, the primer of SI-5 was selected in the following assays.

### 2.5. RPA Primer and sgRNA 

The detailed sequence of *S*. Indiana A7P63_0910 was collected from the NCBI database. According to the conserved sequences of A7P63_0910, Primer Premier software 5.0 was used to design crRNAs and primers. A total of 7 pairs of RPA primers (Table 2) and 10 sgRNAs (Table 3) were included in this study. 

All the primers and sgRNAs were synthesized by Sangon Biotech (Shanghai, China). The Cas12b sgRNA was purified by using Cas12b High Yield sgRNA Synthesis and Purification Kit (31904, ToloBio, Shanghai, China) according to the instruction of the manufacturer.

### 2.6. RPA-CRISPR Cas12b Detection System

The detection process was carried out in one tube and includes two parts: the RPA amplification system and Cas12b detection. As for RPA amplification, 25 μL of suspended mixed reaction buffer V and 2 μL of each primer (10 μM) were used. As for the Cas12b detection system, 1.25 μL of AapCas12b (10 μM), 1.25 μL of Cas12b-crRNA (10 μM), 2.5 μL of HOLMES ssDNA reporter (10 μM), 5 μL of template, 5 μL of magnesium acetate, and 6 μL of nuclease-free water were mixed together to produce a total volume of 50 μL. Then, the dynamic FAM fluorescence signals were simultaneously collected at 37 °C every 30 s for 45 min using the Applied Biosystems QuantStudio 5 real-time PCR system (QuantStudio 5, ThermoFisher, USA).

### 2.7. Sensitivity and Specificity Analysis 

As for sensitivity detection, a serial dilution of the recombinant plasmid containing the target sequence of the *S*. Indiana *A7P63_0910* gene (from 25 copies/test to 200 copies/test by 2-fold intervals) was prepared for the tests. Three replicate experiments were performed for each concentration gradient. 

In specificity experiments, the genome DNA of all pathogens was collected by using commercial TIANamp Bacteria DNA Kit (TIANGEN, Beijing, China) in accordance with the manufacturer’s instructions. Three independent reactions were performed as replicates for each pathogen and were used, and deionized water (DW) served as the no-template control (NTC).

### 2.8. Real Sample Testing 

A total of 109 samples, collected from chicken carcass surfaces, were obtained from three processing points (post-shedding, post-evisceration, and post-chilling) at a chicken slaughter plant (Changzhou, China), during September 2023. The sample collection was performed according to the standard methods recommended by The National Health and Family Planning Commission of the PRC (GB4789.4-2016), which involved rinsing the whole chicken carcass with buffered peptone water (BPW) and then incubation at 37 °C for 18–20 h before use as the pre-enrichment broth. For all the samples, 0.5 mL of the pre-enrichment culture was transferred to 10 mL of selective enrichment (selenite cysteine broth, SC) and incubated at 37 °C for 22–24 h. After selective enrichment, SC broth culture was streaked onto xylose lysine desoxycholate (XLD) agar and incubated at 37 °C overnight. Finally, the isolates were subjected to biochemical identification and serological typing. The genomic DNA was extracted from the selective enrichment fluid using the TIANamp Bacteria DNA Kit for *S*. Indiana RPA-CRISPR/Cas12b assay analysis.

## 3. Results

### 3.1. The Design and Selection of sgRNA for the S. Indiana RPA-CRISPR/Cas12b Assay

For the *S*. Indiana RPA-CRISPR/Cas12b assay, a total of eight Cas12b single guide RNAs (sgRNAs) (Cas12b sg-1 to sg-8, as listed in Table 3) were designed and synthesized based on the amplification products of the primer SI-5. Additionally, a modified RPA-based system was used to assess the performance of these sgRNAs. Among them, sg-8 exhibited superior fluorescence and take-off time compared to the other candidates. Consequently, sg-8 was selected for use in subsequent assays (Figure 2).

### 3.2. Identification of the Optimal Concentrations of Cas12b and sgRNA for the S. Indiana RPA-CRISPR/Cas12b Assay

In order to optimize the performance of the *S*. Indiana RPA-CRISPR/Cas12b assay, the optimal concentrations of Cas12b and sgRNA were determined. Both the Cas12b and sgRNA were initially diluted into 10 μM. Subsequently, different volumes of Cas12b (0.3125/0.625/1.25/2.5 μL) and sgRNA (0.3125/0.625/1.25/2.5 μL) were analyzed. The deionized water (DW) served as the negative control (NTC). As shown in Figure 3, the optimal volumes for Cas12b and sgRNA were 1.25 μL, corresponding to the working concentrations of 250 nM, according to the fluorescence intensity value. Consequently, the concentrations of Cas12b and sgRNA were optimized in 250 nM for the detection reaction.

### 3.3. Identification of the Optimal Temperature for the S. Indiana RPA-CRISPR/Cas12b Assay

Temperature plays a critical role in determining the performance of the detection system. To ascertain the optimal temperature for the reaction, the temperature range was set from 37 to 42 °C, with 1 °C increments. The NTC reaction was set as mentioned above. As shown in Figure 4, The fluorescence intensity of the reaction at 41 °C was much higher than that of other groups. Therefore, the RPA-CRISPR/Cas12b assay was performed at 41 °C in subsequent assays.

### 3.4. Identification of the Optimal ssDNA Reporter for the S. Indiana RPA-CRISPR/Cas12b Assay

The selection of the optimal single-stranded DNA (ssDNA) reporter is crucial for the detection system to accurately identify the target sequence. To determine the most suitable ssDNA reporter, four probes were designed and tested, namely, 8A-FQ (5′-/6-FAM/AAAAAAAA/BHQ1/-3′), 8T-FQ (5′-/6-FAM/TTTTTTTT/BHQ1/-3′), 8C-FQ (5′-/6-FAM/CCCCCCCC/BHQ1/-3′), and 8G-FQ (5′-/6-FAM/GGGGGGGG/BHQ1/-3′. As depicted in Figure 5, only the 8G-FQ probes failed to produce fluorescence signal in the reaction regardless of the presence or absence of the target nuclear acid sequence. Furthermore, the fluorescence intensity of the 8A-FQ probe showed no significant difference between the NTC and test group. Compared with 8T-FQ, the difference of fluorescence intensity between the NTC and test group was much higher in 8C-FQ. Consequently, 8C-FQ was used in the *S*. Indiana RPA-CRISPR/Cas12b assay.

### 3.5. Specificity and Sensitivity of the S. Indiana RPA-CRISPR/Cas12b Assay

To verify the specificity of the *S*. Indiana RPA-CRISPR/Cas12b assay, a total of 22 *Salmonella* strains (including 4 *S*. Indiana strains and 18 other *Salmonella* strains) and 8 non-*Salmonella* strains were collected, seen in Table 1. Deionized water (DW) served as the negative control (NTC), and the recombinant plasmid containing A7P63_0910 served as the positive control (PC). As shown in Figure 6, all the *S*. Indiana strains (4/4) showed positive results, while no signals were detected from other 18 *Salmonella* strains or 8 non-*Salmonella* strains and the negative control. These findings indicated that the specificity of the RPA-CRISPR/Cas12b assay for *S*. Indiana reached 100%, with no false-positive results for other pathogens. Therefore, our present detection system demonstrated a superior specificity for *S*. Indiana.

As for sensitivity, a serial dilution of the recombinant plasmid containing the target sequence of the *S*. Indiana *A7P63_0910* gene (from 25 copies/test to 200 copies/test by 2-fold intervals) was established to examine the sensitivity of the present system. DW was used as the negative control. The limit of detection (LoD) of our method was 14.4 copies per reaction (Figure 7). 

### 3.6. Application of the S. Indiana One-Step RPA-CRISPR/Cas12b Assay to Real Samples

To assess the performance of our present methods for real samples, a total of 109 chicken carcasses were classified by the *S*. Indiana RPA-CRISPR assay and traditional culture method from three processing points (43 post-shedding, 35 post-evisceration, and 31 post-chilling). As shown in Table 4, 17 *S*. Indiana positive samples were identified during the entire processing step, comprising 9 post-shedding, 5 post-evisceration, and 3 post-chilling. The corresponding *S*. Indiana positive rates of post-shedding, post-evisceration, and post-chilling were 20.93% (9/43), 14.29% (5/35), and 9.68% (3/31), respectively. The results obtained from the *S*. Indiana one-step RPA-CRISPR/Cas12b assay were entirely consistent with those by using traditional culture method, achieving 100% consistency with no false-positive or false-negative results observed.

## 4. Discussion

*S*. Indiana is increasingly becoming a major cause of mortality and morbidity in developing countries, exacerbating the burden on global disease [24]. Moreover, the outbreak of *S*. Indiana-induced foodborne disease remains an intractable food safety and public health concern in China [25]. Due to the dynamic antimicrobial resistance and multidrug-resistant of *S*. Indiana, it is difficult to select a useful agent in the treatment of the associated disease [26]. Therefore, the development of an early, rapid, sensitive, and accurate detection tool for *S*. Indiana is of great value in safeguarding consumers from foodborne diseases and ensuring thorough food product safety.

Traditional bacteriological methods for detecting *S*. Indiana are time consuming and laborious, hindering timely surveillance and effective pathogen control. In 2018, our team identified a unique gene, named A7P63_13850 in *S.* Indiana through comparative genomics which is not recorded in other *Salmonella* serovars or any non-*Salmonella* bacteria. Then, a PCR assay was established for specific detection of *S.* Indiana. The detection limit of this method is 10 pg per reaction for bacterial genomic DNA, and the serovar-specificity was verified by bacteriological methods [11]. Although PCR remains the gold standard for monitoring food contamination by pathogens [27], its reliance on sophisticated equipment, skilled technicians, and time-consuming processes limits its applicability in resource-limited regions. In contrast, the isothermal amplification technique presents outstanding advantages, amplifying isothermally at 22–65 °C within 30 min without specific equipment or the requirement of skilled operators [28]. 

The isothermal amplification of nucleic acid technology has been used for the rapid pathogen detection of infectious diseases. Yeun-Jun Chung and co-workers have developed a LAMP assay aiming at *Salmonella* detection, offering an LoD of 20 copies per reaction [29]. In 2021, our group established the loop-mediated isothermal amplification (LAMP) method for rapid detection of *S*. Indiana. The sensitivity is 59 copies per reaction, and the specificity is consistent with GB4789.4-2016 [30]. In 2022, Xiang et al. developed a microfluidic genoserotyping strategy incorporating LAMP for the detection of 11 common *Salmonella* serotypes from retail food samples, including Indiana. The limit of detection was 10^2^ or 10^3^ CFU/mL. Referring to the results of the standard culture method, the LAMP chip was verified with 100% accuracy by testing 688 *Salmonella* and 22 non-*Salmonella* strains [31]. 

Despite the advantages of LAMP due to its low operating cost [32], two significant limitations impede its wider application: (1) the complex 4- or 6-primer system limits the design of multiple target detection schemes; (2) false positives reduce the reliability of testing results due to the high concentration of primers and the complexity of amplicons. Compared with other isothermal amplification technologies, RPA has a fairly simple primer design and can be performed near ambient temperature (37–42 °C). In addition, the RPA assay has been demonstrated to tolerate certain inhibitors [33]. Therefore, RPA methods are an alternative option for pathogen detection in low-resource settings. Recently, the RPA method has been used for the detection of *Salmonella enterica* [15]. Moreover, Yan Jin and colleagues have developed a RPA-based platform for *Salmonella* Typhimurium detection (*S.* Typhimurium), which could work at concentrations as low as 1.0 × 10^2^ copies/µL within 30 min [34]. However, efforts are still needed to eliminate the undesired nonspecific amplification and false-positive results.

To address these issues, RPA integrated with CRISPR/Cas12 (RPA-CRISPR/Cas12) methods has been developed and utilized for detecting pathogens [35,36]. In the present study, we first introduced a method for *S.* Indiana detection via combining RPA with the CRISPR/Cas12b system. Our data suggested that the whole reaction time is shorter than that of traditional PCR method, which was limited to 1 h. Moreover, the reaction temperature of RPA-CRISPR/Cas12 was 37 °C, and the entire mixture can be pre-prepared. Therefore, there was no sophisticated instruments or requirement for specialist technicians. Compared with our previous publication [23], our present reaction can be performed on a simple heat block and carried out by an operator with basic training. Our assay is simpler and more affordable than that of PCR-based methods, making it suitable for resource-limited settings.

Unlike Cas12a, Cas12b has a wide temperature adaptation and works at temperatures from 37 °C to 60 °C, which overlaps the thermal profile of the recombinase enzyme in the RPA assay at 37 °C [37,38]. Moreover, Cas12b barely allows any mismatches between sgRNA and target DNA sequences, endowing higher specificity and accuracy than the CRISPR/Cas12a detection method [39,40]. In the present study, the RPA-CRISPR/Cas12b showed a higher specificity than that of PCR in clinical sample detection. Therefore, the RPA-CRISPR/Cas12b method contributed to reducing the false-positive results in clinical sample detection and presented a superior specificity for *S.* Indiana.

The traditional CRISPR-based methods involve separate amplification of the target gene and CRISPR/Cas detection steps, which may introduce contamination [41,42]. An et al. established a one-tube and two-step reaction system for *Salmonella* spp. detection by combining recombinase polymerase amplification (RPA) with CRISPR-Cas13a cleavage. The detection results of one-tube and two-step RPA-Cas13a and real-time PCR were highly consistent in clinical samples [43]. Compared with Cas12, Cas13 used for nucleic acid detection not only requires reverse transcription, but also the stability of RNA probes is not as good as DNA. Therefore, establishing a CRISPR detection system based on Cas12 offers lower costs and higher stability.

Comparing with the traditional methods, RPA-CRISPR/Cas12b combines the amplification and Cas collateral cleavage in one reaction pot without transferring the amplification products to the Cas reaction system. Therefore, the present methods significantly avoid the contamination, resulting in accurate, sensitive, and rapid methods for *S.* Indiana detection. To overcome the typical drawbacks of this method, the nuclear acid amplification and detection were simultaneously carried out in one step via combining the RPA-CRISPR/Cas12b and Cas collateral cleavage detection system in one pot. Therefore, our assay provides accurate, sensitive, rapid, and contamination-free detection of *S*. Indiana. However, it is important to note that the sample size utilized in the current study is relatively small, and the availability of bacterial strains is limited in chicken carcass surfaces (post-shedding, post-evisceration, and post-chilling). Therefore, the efficacy of the existing detection system warrants further validation across a broader spectrum of clinical isolates encompassing both *Salmonella* and non-*Salmonella* strains, as well as mixtures of various organisms.

## 5. Conclusions

In conclusion, we developed a novel identification system for *S.* Indiana infection diagnosis by integrating RPA and the CRISPR/Cas12b system in a one-pot, one-step approach, achieving simple, rapid, accurate, and contamination-free advantages. Our assay not only provides an alternative approach to *S.* Indiana detection but also offers insight into modifying the methods for foodborne pathogen detection. 

## Figures and Tables

**Figure 1 microorganisms-12-00519-f001:**
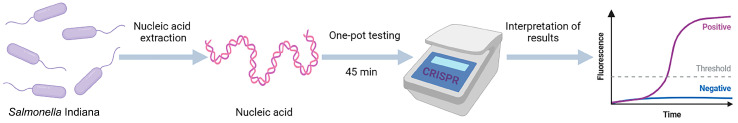
The principle and workflow of the *S.* Indiana RPA-CRISPR/Cas12b assay. Both the working solution of nucleic acid amplification (RPA reaction) and nucleic acid detection (CRISPR/Cas12b assay) were prepared in one tube. The DNA from target samples served as a template and added to the tube directly. Then, the tube was placed on real-time PCR fluorescence readout equipment at a constant temperature. The fluorescence signal can be detected in the presence of positive *S.* Indiana samples, which cannot be observed in negative *S.* Indiana samples.

**Figure 2 microorganisms-12-00519-f002:**
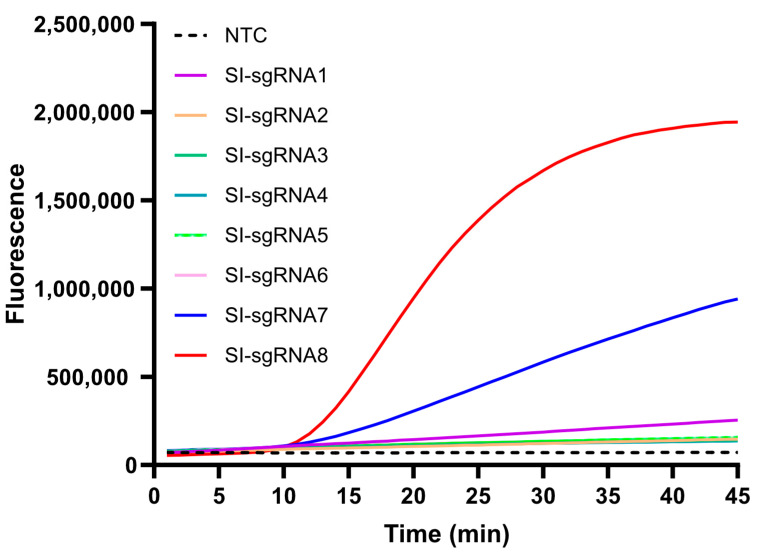
The fluorescence intensity of the reaction by using eight different Cas12b sgRNAs. The fluorescence intensity was close to that of the NTC by using CAS12b sg-2 to sg-6, while sg-8 presented the best performance compared to that of sg-7 and sg-1.

**Figure 3 microorganisms-12-00519-f003:**
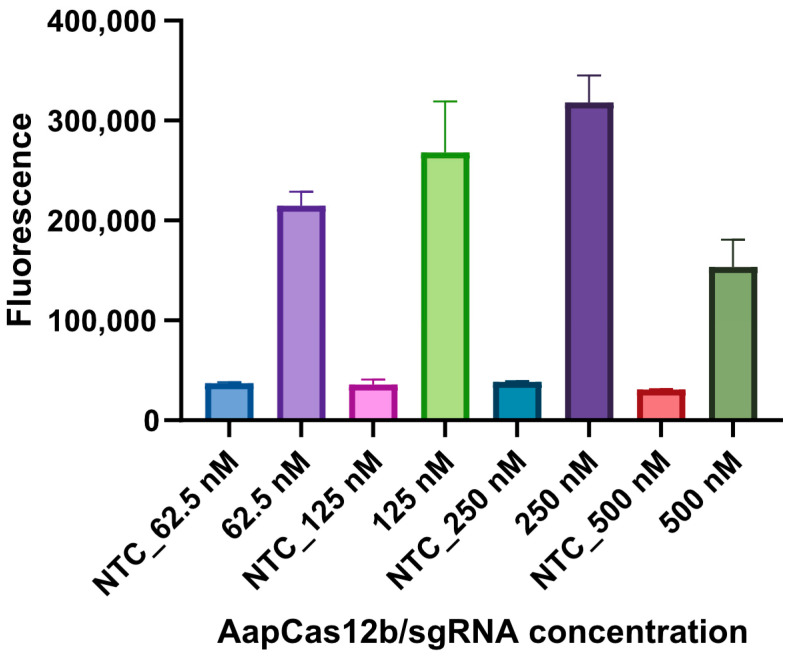
Determination of the optimal concentrations for Cas12b and sgRNA. The fluorescence intensity of NTC reactions with different concentrations of Cas12b and sgRNA (62.5 nM, 125 nM, 250 nM, and 500 nM) showed no significant difference, while the concentrations of Cas12b and sgRNA in 250 nM achieved the highest fluorescence intensity. The recombinant plasmid containing A7P63_0910 served as the template. Three replicates were performed for each reaction.

**Figure 4 microorganisms-12-00519-f004:**
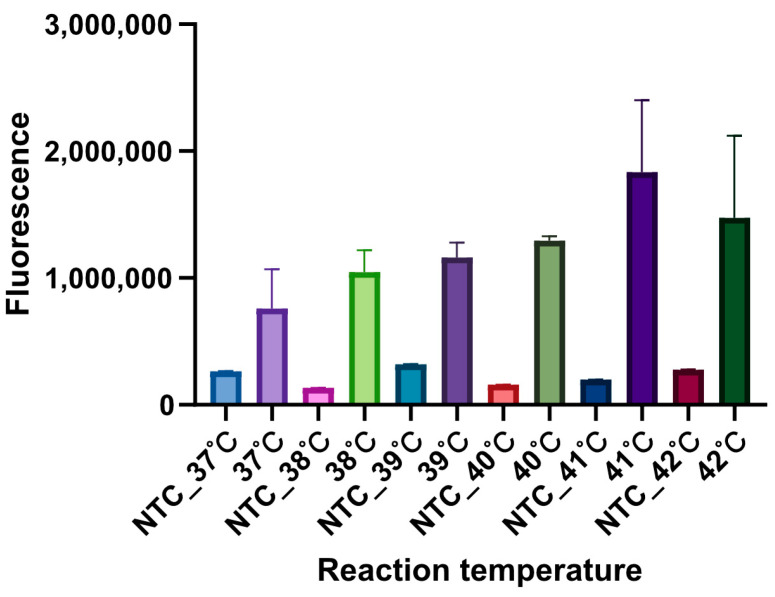
The optimal temperature determination for *S*. Indiana RPA-CRISPR/Cas12b assay. The fluorescence intensity of the NTC reaction showed no significant difference with different reaction temperatures, while the reaction temperature at 41 °C in the test reaction presented the highest fluorescence intensity. The recombinant plasmid containing A7P63_0910 served as the template. Three replicates were performed for each reaction.

**Figure 5 microorganisms-12-00519-f005:**
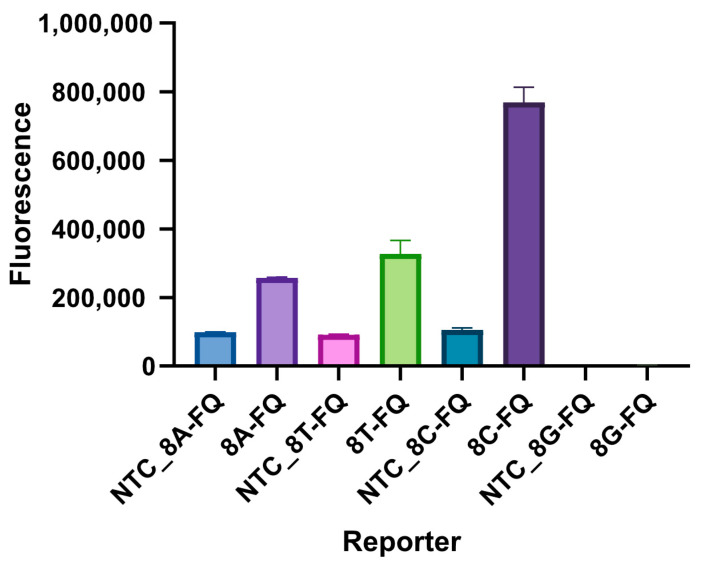
Determination of optimal ssDNA reporter for the *S*. Indiana RPA-CRISPR/Cas12b assay. The fluorescence intensity of the NTC reaction showed no significant difference with different ssDNA reporters (8A-FQ, 8T-FQ, 8C-FQ, and 8G-FQ), while the test reaction by using 8C-FQ ssDNA reporter presented the highest fluorescence intensity. The recombinant plasmid containing A7P63_0910 served as the template. Three replicates were performed for each reaction.

**Figure 6 microorganisms-12-00519-f006:**
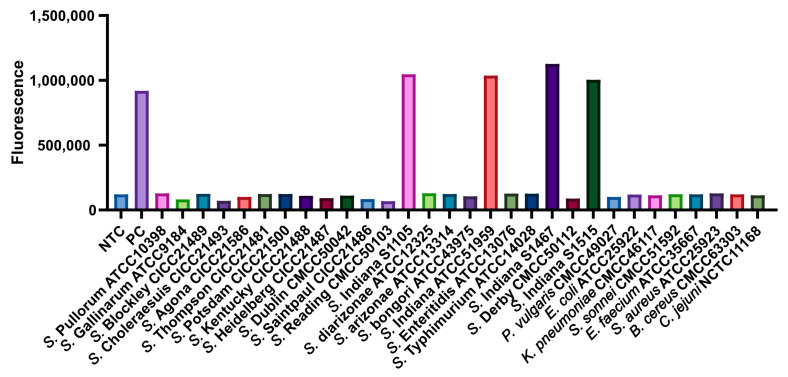
The specificity analysis of the *S*. Indiana RPA-CRISPR/Cas12b assay. Only 4 *S*. Indiana strains and PC reactions exhibited a fluorescence signal, while the fluorescence intensity of other 26 non-*S*. Indiana strains was nearly in the background. Three replicates were performed for each reaction.

**Figure 7 microorganisms-12-00519-f007:**
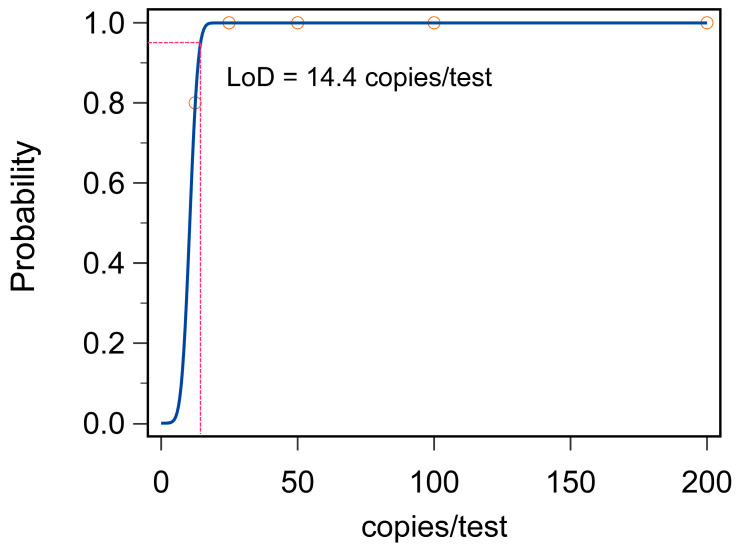
The LoD analysis of the *S*. Indiana RPA-CRISPR/Cas12b assay. The LoD of the *S*. Indiana RPA-CRISPR/Cas12b assay was 14.4 copies per test with the probability over 95%. Three replicates were performed for each reaction.

**Table 1 microorganisms-12-00519-t001:** Strains used for testing the *S.* Indiana RPA-CRISPR/Cas12b assay.

Species		Strain No.
*Salmonella* strains		
1	*Salmonella* Pullorum	ATCC10398
2	*Salmonella* Gallinarum	ATCC9184
3	*Salmonella* Blockley	CICC21489
4	*Salmonella* Choleraesuis	CICC21493
5	*Salmonella* Agona	CICC21586
6	*Salmonella* Thompson	CICC21481
7	*Salmonella* Potsdam	CICC21500
8	*Salmonella* Kentucky	CICC21488
9	*Salmonella* Heidelberg	CICC21487
10	*Salmonella* Dublin	CMCC50042
11	*Salmonella* Saintpaul	CICC21486
12	*Salmonella* Reading	CMCC50103
13	** * Salmonella * ** ** Indiana **	** S1105 **
14	*Salmonella diarizonae*	ATCC12325
15	*Salmonella arizonae*	ATCC13314
16	*Salmonella bongori*	ATCC43975
** 17 **	** * Salmonella * ** ** Indiana **	** ATCC51959 **
18	*Salmonella* Enteritidis	ATCC13076
19	*Salmonella* Typhimurium	ATCC14028
** 20 **	** * Salmonella * ** ** Indiana **	** S1467 **
** 21 **	*Salmonella* Derby	CMCC50112
** 22 **	** * Salmonella * ** ** Indiana **	** S1515 **
Non-*Salmonella* strains		
1	*Proteus vulgaris*	CMCC49027
2	*Escherichia coli*	ATCC25922
3	*Klebsiella pneumoniae*	CMCC46117
4	*Shigella sonnei*	CMCC51592
5	*Enterococcus faecium*	ATCC35667
6	*Staphylococcus aureus*	ATCC25923
7	*Bacillus cereus*	CMCC63303
8	*Campylobacter jejuni*	NCTC11168

Note: The *S.* Indiana strains are in bold font and underlined.

**Table 2 microorganisms-12-00519-t002:** The RPA primers used in this study.

Primer		Sequence (5′-3′)
SI-1	F1	CAGTAGCGACACAATGGAAAATAAATGGAG
	R1	GATTCAGAGTCATATCCCTTACCAGAATCTCC
SI-2	F2	GATATGCAGGGAGATTCTGGTAAGGGATATG
	R2	GTCAAAAACCCTCCAAACATAAACAGTAAACC
SI-3	F3	CAGGGAGATTCTGGTAAGGGATATGACTCTG
	R3	CAGCAAAAAGAGTTGTCAAAAACCCTCCAAAC
SI-4	F4	GGATGTTCTATCTACCACTCGAAAAGAATACG
	R4	CTCCATTTATTTTCCATTGTGTCGCTACTG
SI-5	F5	CGAAAACTCGAAACTACCATGTTTGAATGG
	R5	CCCTTACCAGAATCTCCCTGCATATCATATTC
SI-6	F6	CTGGTAAGGGATATGACTCTGAATCTCAATG
	R6	GAGTTGTCAAAAACCCTCCAAACATAAACAG
SI-7	F7	TTCAATCCTTGCCCGTCGCGGGGCTGTTATCG
	R7	TCATTGCTGTTAAGAACGGAAAGTGTCATTGC

**Table 3 microorganisms-12-00519-t003:** The sgRNA primers used in this study.

sgRNA	Sequence (5′-3′)
sg-1	GUCUAGAGGACAGAAUUUUUCAACGGGUGUGCCAAUGGCCACUUUCCAGGUGGCAAAGCCCGUUGAGCUUCUCAAAUCUGAGAAGUGGCAC**AGUAGCGACACAAUGGAAAA**
sg-2	GUCUAGAGGACAGAAUUUUUCAACGGGUGUGCCAAUGGCCACUUUCCAGGUGGCAAAGCCCGUUGAGCUUCUCAAAUCUGAGAAGUGGCAC**UUUUCCAUUGUGUCGCUACU**
sg-3	GUCUAGAGGACAGAAUUUUUCAACGGGUGUGCCAAUGGCCACUUUCCAGGUGGCAAAGCCCGUUGAGCUUCUCAAAUCUGAGAAGUGGCAC**AUUUAUUUUCCAUUGUGUCG**
sg-4	GUCUAGAGGACAGAAUUUUUCAACGGGUGUGCCAAUGGCCACUUUCCAGGUGGCAAAGCCCGUUGAGCUUCUCAAAUCUGAGAAGUGGCAC**CAUUUAUUUUCCAUUGUGUC**
sg-5	GUCUAGAGGACAGAAUUUUUCAACGGGUGUGCCAAUGGCCACUUUCCAGGUGGCAAAGCCCGUUGAGCUUCUCAAAUCUGAGAAGUGGCAC**CUCCAUUUAUUUUCCAUUGU**
sg-6	GUCUAGAGGACAGAAUUUUUCAACGGGUGUGCCAAUGGCCACUUUCCAGGUGGCAAAGCCCGUUGAGCUUCUCAAAUCUGAGAAGUGGCAC**UUUACUCCAUUUAUUUUCCA**
sg-7	GUCUAGAGGACAGAAUUUUUCAACGGGUGUGCCAAUGGCCACUUUCCAGGUGGCAAAGCCCGUUGAGCUUCUCAAAUCUGAGAAGUGGCAC**CAAGAGCUAUUUACUCCAUU**
sg-8	GUCUAGAGGACAGAAUUUUUCAACGGGUGUGCCAAUGGCCACUUUCCAGGUGGCAAAGCCCGUUGAGCUUCUCAAAUCUGAGAAGUGGCAC**UGUCGCUACUGAAAAUUCAU**

Note: The target sequences are in bold font and underlined.

**Table 4 microorganisms-12-00519-t004:** The comparison of the traditional culture and *S*. Indiana one-step RPA-CRISPR/Cas12b method.

Samples	Num	Positive Sample		Positive Rate	Consistency
		Traditional Culture	RPA-CRISPR/Cas12b		
Post-shedding	43	9	9	20.93%	100%
Post-evisceration	35	5	5	14.29%	100%
Post-chilling	31	3	3	9.68%	100%
Total	109	17	17	15.60%	100%

## Data Availability

Data are contained within the article.

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
