# Peer review of "A One-Pot Convenient RPA-CRISPR-Based Assay for Salmonella enterica Serovar Indiana Detection"

_microorganisms, 2024, doi:10.3390/microorganisms12030519_

Round 1
Reviewer 1 Report
Comments and Suggestions for Authors
The overall composition of the manuscript is good. The paper is scientifically and methodologically accurate. This manuscript will interest many readers.
However, my recommendation is 'Minor Revision'. More detailed comments are given below.
1) Abstract: Italicize the name of salmonella. Check throughout the document
2) Introduction: The authors mention that “However, these tools are highly dependent on professional equipment, experienced operator and long reaction time”. The authors must mention what advantages their method has over other methods. What are the advantages and costs?
3) The authors need to reinforce their focus on why they carry out this work. Before the Materials and Methods section, the authors must indicate which strategies were used to achieve their objectives.
3) Table 1. The authors mention that they used 3 S. Indiana isolates; however, in Table 1 there are marked 4 S. Indiana. Please verify and modify.
4) Mention the origin of strains S1105, S1467, S1515 or add appropriate references.
5) The authors must explain why they used the sequence of S. Indiana A7P63_09100. Furthermore, it is a hypothetical protein.
6) Line 127: reporter(10μM), 5 μL Template, change by: reporter(10 μM), 5 μL template,
7) Line 158: “was shown in in Figure” change by was shown in Figure.
8) Table 2 must be described in the main document before it appears.
Results section:
9) Sections 3.1 and 3.2 should be moved to the Materials and Methods section because they are not results.
10) A short introduction and conclusion in each section must be provided before describing all the experiments and results that support each section. It is to guide the reader to understand the importance of the results. Furthermore, it is suggested that the authors describe their results further.
11) A limitations paragraph should be added indicating that the number of samples is low and that it is recommended to continue the study with a greater number of clinical isolates that include salmonellae and non-salmonellae, and mixtures of different organisms. Is this technique also applied to patients? Or is it only limited to chicken carcass surfaces (post shedding, post evisceration, and post chilling).
12) Homogenize the abbreviations, Lod or LOD. The same for all the abbreaviatures. Verify and homogeneity.
13) The discussion seems like a report instead of a comparison of the results obtained with previous results. This section is requested to be modified.
Conclusions
14) The conclusion does not reflect the art of the work.
15) Finally, the manuscript contains many typographical and grammatical errors. I suggest authors carefully review the entire main document.
Author Response
Dear reviewer, thank you for your comments, please refer to the attachment.

Reviewer 2 Report
Comments and Suggestions for Authors
In your article you described ‘’ I think this belongs in the detection system description and should go in materials and methods’’. Here are my suggestions
Your article summary is longer than normal. Please shorten it and refer to the most important findings of your research.
Lines 36-46: is similar to the introduction of your paper entitled ΄΄ Prevalence and Antimicrobial Resistance of Salmonella enterica Serovar Indiana in China (1984–2016)΄΄, please differentiate it.
Lines 103-113: please give more details about the conditions of keeping the strains
Lines 123-131: Did you use standards for quantification of Cas12; Please provide more information about calibration curves if they were used and in general for the quantification process.
Lines 158-175: I think this belongs in the detection system description and should place it in materials and methods
Line 192-193: place it with the previous paragraph before the table 3 and figure 2
Rewrite all the legends from the figures making them shorter and easy to understand.
In your text correct the S. Indian with the S. indian because the first letter of a specie is always lower and italics
Enrich the conclusion.
Author Response

(The authors gave the same response as above.)

Round 2
Reviewer 2 Report
Comments and Suggestions for Authors
Thank you for your corrections